# Designing Gold Nanoparticles for Precise Glioma Treatment: Challenges and Alternatives

**DOI:** 10.3390/ma17051153

**Published:** 2024-03-01

**Authors:** Cedric Lansangan, Menka Khoobchandani, Ruchit Jain, Serge Rudensky, Christopher C. Perry, Rameshwar Patil

**Affiliations:** 1Division of Cancer Science, Departments of Basic Sciences and Neurosurgery, School of Medicine, Loma Linda University (LLU), 11175 Campus St., Loma Linda, CA 92350, USA; clansangan@students.llu.edu (C.L.); srudensky@llu.edu (S.R.); 2Department of Surgery, Government Medical College, Miraj 416410, India; ruchitusmle16@gmail.com; 3Division of Biochemistry, Department of Basic Sciences, School of Medicine, Loma Linda University (LLU), 11175 Campus St., Loma Linda, CA 92350, USA

**Keywords:** gold nanoparticle, nanomedicine, glioblastoma multiforme, blood–brain barrier, brain tumor treatment, targeted drug delivery, radiation treatment, boron neutron capture therapy

## Abstract

Glioblastoma multiforme (GBM) is a glioma and the most aggressive type of brain tumor with a dismal average survival time, despite the standard of care. One promising alternative therapy is boron neutron capture therapy (BNCT), which is a noninvasive therapy for treating locally invasive malignant tumors, such as glioma. BNCT involves boron-10 isotope capturing neutrons to form boron-11, which then releases radiation directly into tumor cells with minimal damage to healthy tissues. This therapy lacks clinically approved targeted blood–brain-barrier-permeating delivery vehicles for the central nervous system (CNS) entry of therapeutic boron-10. Gold nanoparticles (GNPs) are selective and effective drug-delivery vehicles because of their desirable properties, facile synthesis, and biocompatibility. This review discusses biomedical/therapeutic applications of GNPs as a drug delivery vehicle, with an emphasis on their potential for carrying therapeutic drugs, imaging agents, and GBM-targeting antibodies/peptides for treating glioma. The constraints of GNP therapeutic efficacy and biosafety are discussed.

## 1. Introduction

Nanotechnology is akin to wizardry in the scientific world, dealing with the manipulation and utilization of matter at the nanoscale—generally 1 to 100 nanometers (nm) [1]. This realm is so tiny that it deals with building blocks on an atomic level [2]. The essence of nanotechnology lies in the size and shape of these particles, both of which significantly influence their properties [3,4]. These properties make NPs versatile for a wide range of biomedical applications. The similarity in size between nanoparticles and biological molecules, such as proteins and DNA, as well as bacteria and viruses, has sparked considerable interest in using nanoparticles (NPs) for diverse biomedical applications [5]. In recent decades, nanotechnology has emerged as a highly promising field, with a particular focus on nanoparticles that exhibit distinct optical, electronic, chemical, photoelectrochemical, or magnetic properties [2,6].

NPs can be broadly classified by their category and properties. They are single-component materials, composites, or core–shell forms, and can be classified into three general categories: polymeric, inorganic, and lipid-based [7]. Examples include liposomes, lipid NPs, polymeric NPs, dendrimers, metallic inorganic NPs, quantum dots, carbon nanotubes, and magnetic NPs [7]. Metallic inorganic nanomaterials include gold, silver, palladium, iron, copper, and platinum. Gold nanoparticles (GNPs) are widely used in the biomedical field as therapeutic and imaging agents [8]. The unique optical and thermal properties of GNPs, coupled with their biocompatibility and ease of functionalization, make them valuable tools in biomedical research and clinical applications, including advanced imaging techniques and targeted therapeutic interventions [9,10].

NPs can be designed and synthesized to be biocompatible by optimizing their properties (i.e., size, shape, surface charge, and composition). Moreover, the body’s defense mechanisms might see these particles as invaders and initiate immune responses or trigger inflammation [11]. Concerted efforts are being made to fabricate nanoparticles to improve their targeting and biocompatibility such that they break down safely after efficiently delivering the payload to the target site [12]. The key factor is to engineer the nanoparticles with disease/tissue-targeted ligands like antibodies, peptides, and small molecules for specific drug delivery and diagnostics [12,13].

Nanomedicine has experienced rapid growth in the treatment of various diseases, including brain cancer, lung cancer, breast cancer, cardiovascular disease, and others [14]. These innovative treatments aim to enhance drug bioavailability and absorption time, minimize release time, prevent drug aggregation, and improve drug solubility in the bloodstream [15]. The field of nanomedicine has ushered in a new era of drug delivery by optimizing the therapeutic pathways of active pharmaceutical ingredients encapsulated within nanoparticles.

Cancer encompasses a diverse group characterized by the uncontrolled growth and spread of abnormal cells [16]. It is pervasive and can affect nearly any tissue or organ in the body [17]. The mortality rate of the glioma, Glioblastoma multiforme (GBM), is notably high, and its aggressive nature distinguishes it from many other cancers, including those of the lung, liver, breast, and prostate. The challenges posed by GBM underscore the critical need for continued research and innovative therapeutic strategies to improve the prognosis and quality of life for individuals affected by this formidable malignancy [17,18].

The therapeutic options for GBM are severely limited relative to the challenges posed by the disease. The standard of care, which includes radiation therapy, is associated with side effects. Therefore, the need for more effective and less toxic treatments remains a pressing concern in the scientific and clinical communities. Current and continuous research endeavors hold the promise of uncovering novel strategies to address the complexities of GBM treatment [19,20,21,22,23,24,25]. Researchers are actively engaged in exploring novel strategies that could provide more effective and advanced solutions for managing the complex and aggressive GBM tumor, including boron neutron capture therapy (BNCT). This active exploration of innovative methods, including emerging therapies, targeted interventions, and advancements in medical science, has the goal of improving the outcomes and quality of life for those diagnosed with GBM.

The present review article focuses on a detailed discussion of GNPs, their physicochemical properties, and their applications in the treatment of diseases like cancer, with a focus on gliomas including GBM. Furthermore, we propose an alternative treatment approach for GBM using targeted GNP-based therapy.

## 2. Gold Nanoparticles and Their Physicochemical Properties

Nanoscale materials have been used since ancient times. Nanoparticles, in particular GNPs, and their medicinal applications are very attractive because of their desirable and manipulable physicochemical properties (such as composition, size, shape, and surface chemistry). We give a brief overview and historical record of gold nanoparticles and their biomedical applications.

### 2.1. Overview and History of Nanoparticles and Gold Nanoparticles

The inorganic metal, gold, has long been of interest for its beauty/longevity and its use in pre-industrialized medicine [26,27]. Bulk gold is extremely oxidation-resistant and biocompatible, and thus very much favorable as a nanoparticle constituent [28]. In contrast to bulk or molecular gold, nanoscale gold, when suspended in a solvent such that sub-micron gold particles are homogenously distributed, is called colloidal gold. Preserved documents on the Chinese “golden solution” and Indian “liquid gold” medicines are among the earliest reports on such GNPs [29,30]. However, GNPs became popular based on the respective seminal work of both the English scientist Michael Faraday and the German physicist Gustav Mie. Faraday demonstrated the range of brilliant colors of GNP colloidal solutions, while Mie provided mathematical solutions to Maxwell’s equations that describe absorption and scattering of GNPs [31]. Since then, many synthesis strategies, shapes, and biomedical/industrial/optical applications for GNPs have been developed, including the Turkevich, Frens, and Schiffrin–Brust methods. Figure 1 shows an illustrative view of the most common shapes/types of GNPs and their surface modifications.

### 2.2. GNP Shapes

Although GNPs come in many shapes and sizes, the following are the most common shapes of GNPs studied in the literature:Nanospheres;Nanorods;Nanostars;Nanocubes;Nanotriangles/Nanopyramids.

Most biomedical applications of GNPs for cancer therapeutic research possesses some variation in these shapes [32]. Of the five shapes listed here, nanospheres are the most commonly used. Solution-based synthesis via chloroauric acid reduction is the most popular way to make GNPs in the range of 5 to 100+ nm in diameter [33]. Nanospheres have historically been most used for imaging and radiation dose enhancement applications. They are also related to nanoshells, which have a silica core surrounded by a thin layer shell of gold, hence their name. Given that nanoshell thickness and core diameter can be adjusted, their corresponding optical properties can also be similarly tuned. Furthermore, hollow nanospheres have been employed elsewhere because of their excellent plasmonic photothermal activities (detailed in the next section) [33]. The spherical shape of nanospheres lends itself to highly efficient ligand packing. A vital part of ligand functionalization is the formation of a monolayer on these nanospheres, which in the case of thiol ligands has been shown to lead to self-assembling monolayer (SAM) coverage [33,34]. In another study, cellular uptake of nanorods and nanospheres in HeLa cells was inversely proportional to their aspect ratio, indicating that this cell line took up nanospheres more than nanorods [35].

Nanorod synthesis via a gold seed-mediated growth method (see Section 4.1) also often involves the cationic surfactant cetyltrimethylammonium bromide (CTAB) acting as a stabilizing agent [33]. These GNPs have been used to conjugate a broad spectrum of materials to yield novel and improved plasmonic properties [36]. Nanorods have at least two localized surface plasmon resonance (LSPR) bands, (1. Transverse, and 2. Longitudinal) due to the anisotropic shape (i.e., length and width) of these GNPs (see Section 2.3.1 for more) [37].

The other three shapes, namely nanostars, nanocubes, and nanopyramids (or nanotriangles) are gaining in popularity. Gold nanostars, which are anisotropic with multiple branches from the particle core, are known for their greater aspect ratio and thus higher surface area and reactivity compared to their counterparts [38]. This high reactivity is attractive for photothermal therapy applications for cancer [39]. Gold nanocubes, meanwhile, can serve as building blocks for plasmonic applications, particularly because of the LSPR foci at their vertices and edges (refer to Section 2.3.1 for details on LSPR) [40]. Unlike nanorods, their CTAB layer is thick and thus much more difficult to replace by ligand exchange with citrate. For similar reasons as both nanostars and nanocubes, nanopyramids/nanotriangles, by virtue of their anisotropic nature and surface area afforded by their shape, are increasingly looked at as dual photodynamic and photothermal therapy agents for cancer [39,41]. Campu and colleagues showed great preclinical promise in terms of melanoma cell death before and after irradiation for their dual photothermal and photodynamic therapy treatment [41].

### 2.3. Physicochemical Properties of GNPs

Many aspects of GNPs, as illustrated in Figure 1, also affect their properties, some of which are crucial determinants of biocompatibility, biodistribution, and cytotoxicity. The physicochemical properties of GNPs include size, shape, coating, functional group/ligand attached, and/or surface charge (zeta potential). These properties enable high drug-loading capacity (via their immense surface area), storage stability (i.e., colloidal stability), and adaptability in the mode of administration [42]. Because of their facile and well-defined synthesis and functionalization methods, GNP formulations of any combination of the above are theoretically possible for virtually any desired application while importantly remaining biocompatible and non-toxic.

#### 2.3.1. How GNP Size and Shape Affect Their Physicochemical Properties

GNP size is extremely important for not only determining optical properties, but also for its renal clearance (and by extension its blood half-life and whole-body half-life). NPs less than 10 nm in diameter are easily filtered by the kidneys, and those larger than 200 nm are removed by the reticuloendothelial system (phagocytosing cells) [43]. Other sources of NP removal from circulation also involve phagocytosis or similar filtration activity in the liver or spleen. Thus, size is a vital determinant of retention time and bioavailability, especially since NPs less than 100 nm demonstrate longer circulation periods in the bloodstream before being cleared. NPs in general are very prone to both aggregation and immune opsonization, the latter involving the deposition of markers (usually proteins) that signal phagocytes to engulf and eventually eliminate pathogens and foreign substances. These interactions of NPs with serum proteins form what has been dubbed the protein corona because of the crown-like arrangement of serum proteins that bind to the surface of NPs [43,44,45]. Formation of the protein corona adds to the “base” diameter of the NP, leading to a separate but related and physiologically relevant size called the hydrodynamic diameter (HD) [46]. Since globular proteins with an HD of 5–6 nm are known to be renally filtered into the urine, this size range and below (more specifically, an HD ≤ 5.5 nm) is associated with being the renal clearance limit [47].

The shape of GNPs also governs their physicochemical and optical properties [48,49]. Optical properties are particularly dependent on GNP shape [38,50]. Mie’s pioneering work building off Faraday’s findings led to the use of Maxwell’s equations for solving the absorption and scattering of electromagnetic radiation (i.e., light) by the nanospheres in the colloidal solution [51]. The vibrant and variable colors of GNPs are thought to be created by the enormous amount of light absorption (i.e., extinction) that occurs during oscillation of the gold atom’s conduction band electrons when they encounter an electromagnetic field (i.e., light) [52]. These oscillations produce surface plasmons (a collective “resonance” of oscillations of the conduction band electrons) in a phenomenon known as localized surface plasmon resonance (LSPR). This LSPR is unique to nanoscale materials and thus does not occur with bulk gold. LSPR for GNPs depends on their size, shape, dielectric properties, aggregation, surface modifications, and medium refractive index [51,53].

Visible light absorption spectra of GNPs will exhibit a characteristic strong LSPR peak around 520 nm, but this peak position shifts based on GNP size and surface chemistry [54]. The LSPR of nanostars can be adjusted from the visible to the near-infrared region via size and branching adjustments, thus posing highly favorable customization for biomedical applications [38]. The optical properties of nanospheres can be tuned in the visible region as desired by modulating their synthesized size [51]. Another important shape-dependent optical property of GNPs is surface-enhanced Raman scattering (SERS), in which the focalization of a localized SPR (LSPR) on GNP surfaces leads to enhanced Raman scattering of light at these focal points, producing a unique and highly enhanced light scattering profile and local electric field, thus increasing the Raman signal of molecules nearby the GNP [51,54,55]. In particular, nanorods and nanostars have been extensively studied as GNP shapes for biosensing, owing to their respective LSPR foci at the nanorod surface and nanostar tips. Interestingly, these LSPRs mediate GNP applications in the biomedical field since their enhanced electric fields can be transformed into heat in light-stimulated drug delivery and photothermal therapies [33,56].

#### 2.3.2. Surface Charge and GNP Stability 

The high surface area of GNPs is important for their resistance to agglomeration. Agglomeration refers to the undesirable phenomenon of particles clumping together, which can affect their dispersion and performance in various applications. To understand and evaluate the potential for agglomeration, scientists often use zeta potential analysis. Zeta potential is a measure of the electric charge around the nanoparticles in a solution. It provides valuable information about the physical stability of the particles in the solution. A highly positive or negative zeta potential value indicates a strong repulsion between particles, making them less likely to agglomerate. When particles repel each other due to a highly positive or negative zeta potential, the suspension remains colloidally stable such that the individual particles remain dispersed rather than forming clusters/aggregates. The general criterion for determining the colloidal stability of NPs is often set at ±30 mV. If the zeta potential of a particle suspension is either greater than +30 mV or more electronegative than −30 mV, it is considered extremely stable [57,58,59,60,61]. Between these general ranges (i.e., +30 mV down to 0 mV, as well as 0 mV to −30 mV), the likelihood of agglomeration or flocculation increases as surface charge approaches 0 mV. Nonetheless, GNPs have been synthesized with high monodispersity and improved colloidal stability despite their zeta potentials being out of this range, mostly due to steric factors, electrostatic repulsion, or else chemical stabilization agents conjugated to GNPs as ligands or coatings (see Section 4 for further discussion) [58,59,62,63,64,65]. It is important to maintain colloidal stability in various industrial applications, such as pharmaceuticals, cosmetics, and materials science. Researchers and engineers aim to design and produce nanoparticles with characteristics ensuring their stability and effective performance in various environments [10,57]. 

## 3. Physicochemical Characterization Techniques for GNPs

Characterization techniques are used to determine the composition of synthesized nanomaterials. There are a variety of analytical methods that have been successfully applied to the characterization of nanomaterials. Typically, spectroscopic and scattering characterization techniques span the electromagnetic spectrum and are key to confirming the physicochemical properties desired in GNP preparations (SPR/LSPR, core size, and hydrodynamic size). These methods can be sub-divided into spectroscopic, scattering, and microscopy techniques with origins in the field of colloidal chemistry. Figure 2 summarizes these characterization techniques.

### 3.1. Ultraviolet-Visible (UV-Vis) Spectrophotometry

Ultraviolet-visible spectrophotometry is an optical technique that measures the interaction of electromagnetic radiation with a sample to produce a wavelength-dependent extinction (absorption + scattering) spectrum. Noble metal nanoparticles, especially gold and silver, exhibit strong absorption bands in the visible region known as LSPRs. The LSPR bands are strongly dependent on GNP size, shape, and composition [66,67,68,69]. These properties and others influence important aspects of GNPs, including concentration, size, and aggregation state (Figure 2A) [70,71]. Importantly, gold nanostars and nanorods both exhibit characteristic and strongly shape-dependent dual UV-vis LSPR bands due to the former’s core and spikes [72,73,74], as well as the latter’s transverse and longitudinal particle dimensions/plasmon foci [75]. This dual band is not present in spectra of gold nanospheres, which instead demonstrate a single SPR band around 510–520 nm, depending on their size [10]. Since these properties are important in characterizing GNPs to ensure desired plasmon peaks, UV-vis is widely used as a basic tool to characterize GNPs. 

### 3.2. Dynamic Light Scattering (DLS)

Dynamic Light Scattering (DLS) is a light scattering technique to determine hydrodynamic size (NP and solvent cage) (Figure 2B). DLS has the advantage of determining an ensemble average of NP sizes and relative polydispersity [76]. In this technique, the intensity fluctuations in the scattered laser light (Tyndall effect) due to the Brownian motion of the GNPs are used to determine the particle diffusion coefficient, which is then related to its hydrodynamic radius [76,77]. Brownian motion is the random walk/motion of colloidal particles because of collisions with water or other solvent molecules in solution. The hydrodynamic radius will increase in aqueous biological media (and hence so too in physiological conditions as well as in the body) because of protein, particularly serum, adsorption on GNPs (see Section 2.3.1 for more discussion).

### 3.3. Atomic Force Microscopy (AFM)

Atomic Force Microscopy (AFM) is a scanning probe technique that can be completed under ambient conditions [78,79]. AFM can be used to image matter at the atomic scale through raster scanning the instrument stylus over its surface. The contouring of a sample under ambient conditions with the molecularly sharp stylus in this technique also allows measuring their physical as well as chemical and biological properties via the reflection of an applied laser beam by a sample. The vertical distance between the stylus and the sample as a consequence of the force applied by the stylus on the sample is used to reconstruct the sample’s topography [79]. Figure 2C shows an example of GNP core size topographic measurements using AFM. As part of the big-picture goal of GNP characterization for confirming desired physicochemical properties, this technique can be used to corroborate UV-vis and transmission electron microscopy measurements of GNP core size, albeit with the variables of context either in-buffer or in-air.

### 3.4. Transmission Electron Microscopy (TEM)

GNP core size can also be measured using transmission electron microscopy (TEM) [80]. TEM imaging depends in principle on differential electron absorption and scattering by thinly sectioned plastic-embedded samples. The contrast it provides in sample imaging because of the short wavelength of the electron beam is part of why the technique can achieve a sub-nanometer resolution. TEM is best performed with conducting or semiconducting samples under a high vacuum (10^−7^ torr). The direct imaging capability of TEM is particularly useful for nanomaterials with non-spherical shapes since it allows for direct measurement of the aspect ratio. Figure 2D–F shows examples of TEM images captured of three GNP shapes (nanospheres, nanostars, and nanorods). 

## 4. GNP Synthesis Strategies

GNPs can be prepared using so-called top-down and bottom-up approaches [81]. As its name implies, the top-down method involves breaking apart bulk gold until the desired dimensions are achieved, usually through assembling and forming via a pre-made pattern or matrix. However, the final GNP size and shape are not able to be finely tuned using this method. In comparison, the bottom-up method starts with gold ions and subjects the solution phase to either chemical or biological reduction into single atoms of gold [82]. We will focus on chemical reduction for this review.

### 4.1. GNP Synthesis by Chemical Au(III) Reduction

Reduction in GNP synthesis is a two-step process consisting of nucleation followed by growth. Nucleation is essentially the formation/coalescence of Au(0) into nuclei. Initiated by the reduction of Au(III) to Au(0) by a reducing agent (i.e., borohydrides, aminoboranes, hydrazine, or citric and oxalic acids, to name a few), nucleation does not proceed until the Au(0) concentration surpasses a solubility threshold, at which point nuclei rapidly form/coalesce. Nucleation ends when that critical threshold is no longer met as the Au(0) concentration decreases over time. Nucleation requires the presence of an electrostatic or steric stabilization agent (i.e., surfactant), such as citrate, the cationic surfactant cetyltrimethylammonium bromide (CTAB), or else a carboxylic acid, amine, or thiol [83]. Once Au(0) nuclei form, a growth step then occurs, where more Au(0) nuclei and Au(0) itself further coalesce to form the final GNPs as a colloidal solution. Importantly, the growth step must have a passivator/stabilizer present (also known as a capping agent), which similarly to the nucleation step acts as an electrostatic/steric stabilizer for mediating colloidal stability (i.e., preventing aggregation of GNPs via either electrostatic or steric stabilization). If both steps are performed in one pot, this is called in situ synthesis, or else seed-mediated growth if not [81]. In the latter case, in situ synthesized GNPs are used as seeds for seed-mediated growth, where the GNPs grow larger and larger step by step so that size control is optimized. This two-step process first involves synthesis of GNP seeds followed by the second growth step in a solution containing Au(III) salt along with passivating and reducing agents. Here, the reduced Au(0) deposits and grows on the seeds to ultimately produce larger GNPs as desired. Importantly, the reducing agents for the growth step must be mild so that Au(0) is only deposited on the surface of existing seeds while preventing any new nucleation events [81].

### 4.2. Common Methods for GNP Synthesis

There are three main GNP synthesis methods, some variation or combination of which is a part of most of the GNP research articles that we have found. They all rely in principle on a similar chemical reduction mechanism. The first and most popular of these synthesis methods was developed by its namesake, John Turkevich and colleagues [81,84]. Here, trisodium citrate dihydrate is quickly added with stirring to boiling gold Au(III) salt solution, after which a wine-red colloidal GNP solution of the order of 20 nm should be produced. A second method involves a modification of the Turkevich method developed by Frens to modulate the trisodium citrate:Au(III) ratio to synthesize 15–150 nm particles, though 20+ nm syntheses using it are polydisperse [85]. The third common synthesis method is known as the Brust–Schiffrin method, which was first to result in thiolate-stabilized GNPs as an in situ synthesis [81,86]. This approach has evolved over the years from a two-phase synthesis to a much simpler method with several upsides compared to the Turkevich method. Its advantages include ambient reaction temperature, excellent thermal and air stability, low aggregation or decomposition, sub-5 nm size range (2–5 nm) with high monodispersity, and simple functionalization/modification methods via ligand substitution. Because of the gold-S bonds, the GNP stability is relatively high, and the resulting shapes are cuboctahedral and icosahedral. Here, smaller GNP core sizes can be achieved by increasing the sulfur:gold (S/Au) molar ratios.

## 5. GNP Applications in Cancer Nanotherapeutics

Several lines of recent biomedical research have utilized GNPs. The properties of the GNPs used in these studies impacted their corresponding biological efficacy in various ways and will be discussed. Cancer treatment applications of GNPs are provided. Table 1 summarizes the applications of GNPs for various cancers.

GNPs are increasingly gaining popularity because of their biological efficacy in cancer therapeutic applications/development [48,96,97,98]. Five cancers which have recently documented applications in a research context are breast, colon, lung, prostate, and pancreatic cancers [36]. The MCF-7 breast cancer cell line treated with GNPs conjugated with extract from the Anacardiaceae showed greater dose-dependent cytotoxicity compared to control [99]. Another study targeted GNPs to breast cancer cell lines using CD44 as a ligand conjugate and found high cancer-cell-specific accumulation and reduced tumor growth in vivo [100]. Furthermore, GNPs loaded with a radiolabeled version of the antioxidative polyphenol Resveratrol were used to treat HT29 colon cancer cells both in vitro and in vivo, resulting in significantly reduced cancer cell viability [101]. Moreover, the treatment of colon cancer cell lines with GNPs electrostatically conjugated to the candidate cytotoxic compound SN38 resulted in high red-LED-stimulated drug release and cytotoxicity [102]. As for lung cancer, the treatment of lung cancer cell lines with GNPs loaded with anti-cancer docetaxel and tumor-targeting folic acid resulted in highly specific and effective cancer cell death [103]. Wang and colleagues loaded GNP nanocages with the anti-cancer paclitaxel and a micelle for green synthesis of GNPs successfully induced tumor cell death and dysfunction [104]. Lastly, chitosan-capped GNP nanorods loaded with gemcitabine and coated with silica were fabricated by another group to treat pancreatic cancer cell lines via acidic-pH-sensitive drug release, finding greater cytotoxicity than with drugs alone [105].

## 6. GNPs as a Drug Delivery Vehicle

Systemic drug delivery should ideally be targeted and specific to avoid the issues with diffusion and off-target side effects as discussed earlier. While this review is focused on cancer applications of GNPs, there are also several examples in the literature of GNP systemic delivery of therapeutics which have been demonstrated to show efficacy [106,107,108]. However, these untargeted therapeutics rely on passive accumulation in diseased tissues in vivo through the leakiness that occurs in the surrounding vasculature, thus posing a risk of systemic toxicity and other off-target effects. Indeed, despite their favorable properties and great potential for targeted drug delivery, GNPs for this purpose also face several challenges and limitations. For example, a central concern we will briefly focus on here is that uncoated and citrate-stabilized GNPs are prone to aggregation in physiological conditions. This is due to not only formation of the protein corona, but also the related “wild cards” which the bloodstream and local tissues present to those in circulation, particularly the neutralization of charges caused by the ionic strength of physiological salts/electrolytes [101,108,109]. Furthermore, GNP synthesis as outlined in Section 4 often involves chemicals, compounds, and polymers which, if not purified out completely, present major toxicity concerns [110,111,112,113,114,115,116]. Given these challenges and limitations, more of which are discussed in Section 6.1 below, here, targeted brain and central nervous system delivery via GNPs is surveyed and discussed with respect to their specific in vitro and in vivo accumulation and bioefficacy. Please refer to Table 2 in Section 9 for a summary of a selection of the work and findings discussed in this section.

### 6.1. Strategies and Examples of GNP Drug Delivery in the Brain

Barriers to effective drug delivery in the brain are manifold, but the most important and pervasive is the blood–brain barrier (BBB) [117]. Made mainly of tightly packed cerebral capillary endothelial cells (CCECs), the BBB is flanked by pericytes and presynaptic astrocytes through the basement membrane [118]. This multilayered barrier, though semipermeable, effectively blocks the free movement of molecules (and particularly toxic chemicals) between the blood and brain parenchyma, which significantly limits the entry of therapeutics to the brain. Schematic representations of the BBB are shown in the figure in Section 8.

Most NP-based drug delivery mechanisms for the brain utilize one of three pathways: passive diffusion, carrier-mediated influx, and vesicular transport including receptor-mediated transport [119]. Other mechanisms are either too invasive and carry major risks associated with them (e.g., neurosurgery) or else too limited to lipid-soluble small molecule delivery. Other means of penetrating the BBB include intrathecal injection, osmotic destruction, ultrasound or magnetic interference, and nasal/olfactory administration, though they have detrimental effects on the BBB and biosafety in general [119]. Wheat germ agglutinin linked to horse radish peroxidase (WGA-HRP) has been previously shown to mediate absorptive endocytic transport across the olfactory cerebrospinal fluid (CSF) barrier via WGA binding neural membrane lectin sites [120].

Zhang and colleagues synthesized WGA-HRP conjugated to GNPs as targeting and imaging moieties to improve central nervous system delivery of therapeutic drugs [119]. The targeted GNPs were loaded with several polymer-linked drugs for the treatment of spinal-cord-injury-induced diaphragm paralysis [119]. This system takes advantage of the WGA lectin recognized by N-acetylglucosamine and sialic acid receptors found on neuronal cell membranes. The specific accumulation of the GNP-delivered respiratory-neuron-targeted therapeutic drugs in the cervical spinal cord and medulla would be vital to avoid the detrimental side-effects of systemic non-targeted administration (i.e., nausea and seizures associated with drug diffusion and its resulting off-target effects). Importantly, a biodegradable bond between the drugs and the GNP mediated drug release at the target tissue. The final three GNP nanoconjugates (two conjugated to the prodrugs before intracellular activation, and one with only WGA-HRP-GNP) were evaluated by a single injection of each into the diaphragm muscles of separate C2Hx model rats with induced left hemidiaphragm paralysis (confirmed by electromyography, or EMG). All the rats treated with the drug-loaded WGA-HRP-GNPs exhibited a restoration of diaphragm muscle function as evaluated by EMG. Immunohistochemical analyses confirmed the specific as-targeted accumulation of the GNPs at the tissue sites.

Although nanocluster-size GNPs could benefit from passive diffusion mechanisms, larger GNPs must be functionalized to utilize an active transport pathway [117]. Nanocluster GNPs of the size 0.9–1.5 nm can cross the BBB directly through ion channels of the same order of dimension, although their high concentration can exert osmotic pressure or else depend on transient changes in BBB membrane fluidity sufficient to allow GNP penetration [121,122]. A study by Gao and colleagues synthesized GNP nanoclusters for the delivery of novel anti-Parkinson’s Disease (PD) drugs [123]. Their N-isobutyryl-L-cysteine (L-NIBC) protected GNP nanoclusters administered to both in vitro PD cell line models and a murine PD model effectively ameliorated alpha-synuclein (alpha-Syn) fibrillation and reduced behavioral disorder phenotypes. Importantly, GNP nanocluster treatment in their study was shown to reverse dopaminergic neuron loss and decrease tyrosine hydroxylase expression in their chemically induced PD animal model. Regarding larger GNPs, several receptors known to be overexpressed on the membranes of the BBB have been leveraged to functionalize GNPs for receptor-mediated endocytosis entry. These receptors include those for transferrin, low-density lipoprotein, and insulin. Moreover, electrostatic interactions have been used to maximize the entry of positively charged GNPs (via their coating and/or ligands) into anionic CCECs [117].

## 7. Gliomas, Glioblastoma Multiforme (GBM), and the Promise of Nanoparticles

### 7.1. Glioblastoma Multiforme (GBM) Overview

Glioblastoma multiforme (GBM) is a grade IV glioma arising from glial and neuronal cells. It is the most common malignant tumor in the central nervous system [124]. The location of GBM is dependent upon the amount of white matter present. The majority of GBM cases are supratentorial (31% temporal, 24% parietal, 23% frontal, 16% occipital) which is about more than 75% of tumors, while <20% are multifocal and about 2% to 7% are multicentric [17,125]. GBM accounts for nearly 80% of all malignant brain tumors and occurs with an incidence of 3–6.4 cases per 100,000 people annually. Nearly 13,000 new cases of GBM are diagnosed annually in the United States alone [125]. Despite the standard of care, the mean overall survival period from diagnosis is less than 20 months. Moreover, the 5 year survival rate is only about 5.8% after standard of care treatment [16]. GBMs are highly diffusive and infiltrative in nature, often growing a few centimeters from the core, thus making it extremely difficult to achieve complete remission.

For most of the patients with GBM, there are no known causative factors. Few risk factors have been determined to date to be linked to GBM tumorigenesis. They include age (incidence increases with age such that it peaks at 85 years and decreases after 85 years), environmental factors (such as DNA damage or dysregulation due to exposure to ionizing radiation, as well as tobacco smoking exposing users to BBB-penetrating nitrosamines and aromatic hydrocarbons), nutritional risk factors (such as obesity), pesticide exposure, familial history, and many more. Some studies also suggested the effect of ovarian steroid hormones in the development of GBM. Interestingly, allergy and infection appear to be protective against GBM [17,126,127]. Antihistamines have been shown to reduce the risk of developing gliomas. Aspirin and its metabolites have been shown to reduce the risk of GBM by blocking the attachment of fibroblast growth factor (FGF) to its receptor and thus inhibiting the growth of GBM cells. Moreover, cannabinoids have demonstrated tumor volume reduction effects in animal models. Various genes have been identified to contribute to tumorigenesis, including MGMT gene with promoter methylation, IDH1 gene mutation, EGFRv3 gene variant, BRAF gene V600E mutation, and ATRX [17,125].

WHO has classified GBM by two different categories. One category is by cell type and includes Astrocytoma, Glioblastoma and Oligodendroglioma, while the other category is by malignancy grade as WHO Stages I–IV. In 2021, The WHO CNS5 took a new approach and classified Glioneuronal tumors into six families: (1) Adult-type diffuse gliomas, (2) pediatric-type diffuse low-grade gliomas, (3) pediatric-type diffuse high-grade gliomas, (4) circumscribed astrocytic gliomas, (5) Glioneuronal and neuronal tumors, and (6) Ependymomas [128,129].

GBM patients present with a variety of neurological symptoms, the most common of which are headaches (seen in about 50% of patients). Papilledema is seen in patients with increased intracranial pressure (ICP) due to tumor volume. Cognitive difficulties and personality changes may occur, depending on the location of the tumor. Some may present gait imbalances and incontinence. A few cases of GBM present so abruptly that they may resemble a stroke. Various modalities are used to diagnose GBM, with magnetic resonance imaging (MRI) being the gold standard for diagnostic imaging. Computed tomography (CT) also provides very useful information and is reserved for patients who are not able to undergo an MRI like patients with pacemakers. Patients who show positive enhancement with gadolinium contrast with an area of central necrosis surrounded by white matter edema. Tumors can be unifocal and multifocal, and can mimic various non-cancerous lesions in the brain such as brain abscess, stroke or multiple sclerosis (MS). The diagnosis of gliomas has been hindered by various obstacles, including tumor heterogeneity, the BBB, and tumor complexity [17,128,130].

### 7.2. Current Treatment Modalities and Their Limitations

Various modalities used to treat GBM are shown in Figure 3. Developed by Stupp and colleagues in 2005, the gold-standard standard of care for GBM patients involves maximal surgical resection (surgical removal of as much of the tumor as possible (also known as tumor debulking)) followed by combined radio-chemotherapy (photon/proton irradiation of the tumor cavity and concomitant treatment with the alkylating chemotherapeutic drug temozolomide, TMZ) [131,132]. Various tools have been implemented to improve the safety of resection such as intra-operative ultrasound/MRI and functional mapping. Contrast-enhanced MRI 72 h before the procedure increases the chances of evaluation for the surgical resection. The presumed progression of tumor cells along neuronal fibers without macroscopic changes leads to a failure to achieve clear surgical margins [17,125]. However, this clinical standard and the multitudes of novel treatments currently in or that have gone through clinical trials for safety and efficacy face several hurdles, including tumor heterogeneity, the BBB, and tumor complexity [124]. One experimental therapy which cleared trials and is now included in the standard of care is the use of tumor-treating fields, which have improved survival in GBM patients [129].

Despite the standard of care and decades of research into alternative therapies, GBM patient prognosis remains dismal [126]. Factors which are associated with poor prognosis include low pre-operative Karnofsky performance status, age, tumor location, and the presence of molecular factors like methylation of the MGMT gene promoter and mutation of the IDH-1 gene. The extent of resection is one of the most important prognostic factors since GBM is characterized by diffuse infiltration into brain parenchyma (functional tissue). Thus, resection success is directly related to increasing overall survival and prolonging time until the inevitable tumor relapse [128]. Though the scientific community has made new discoveries and treatment advancement, GBM still remains an incurable cancer. The goal of medical management is to make a diagnosis and to prolong and improve the quality of life of the patient [17].

Compared to other malignant tumors, the BBB presents a unique and challenging barrier to effective drug delivery for treating GBM. Aggravating the challenge is that the BBB, despite increased angiogenesis throughout GBM progression, still does not permit the entry of most drugs via the enhanced permeability and retention (EPR) effect [117]. Thus, chemotherapeutic drugs which are commonly used are not effective in GBM due its poor penetration across the BBB. All these factors lead to poor prognosis and tumor recurrence [133,134]. That said, in 2002, the FDA approved the nitrosourea-containing BCNU (carmustine) polymer wafers (marketed as Gliadel). These carmustine-containing polymer wafers deliver the alkylating agent, which crosslinks DNA and thus blocks replication and transcription [135]. Xing and colleagues reviewed the literature in 2015 to determine if GBM overall survival significantly improved with carmustine wafer treatment of newly diagnosed GBM, finding that there was no statistically significant increase in survival for the treatment in two randomized clinical trials compared to control. However, the four cohort studies also examined in the review did exhibit a significantly increased survival with treatment [136]. Approximately 1600 clinical trials are currently categorized as “Glioblastoma” on “ClinicalTrials.gov”. Despite decades of such research, the long-term prospects for patients with GBM remain extremely poor. Currently, no screening tool or test is available for detecting GBM in early stages before the onset of symptoms [17,137].

### 7.3. Potential Treatment Modalities for Glioma

Numerous novel treatments for GBM have been investigated. So-called GBM stem cells (GSCs) located in the perivascular and hypoxic zones of the tumor are believed to underlie tumor initiation, radiochemoresistance, and recurrence/relapse [138,139]. Targeting these GSCs through their unique stem-like surface markers in conjunction with other treatments and the standard of care pose an exciting area of research currently being pursued [140]. Immune system therapies have also been a hot-topic research item, though none have yet yielded a clinically approved treatment. These include immune checkpoint blockade therapies, the co-administration of convection-enhanced delivery of TMZ and whole-cell immunization with irradiation-inactivated tumor cells, sex-specific differences in myeloid cell populations in GBM murine models leading to differential immunosuppressive properties, and immunization with specific virus antigens to elicit local immune activation and instigate antineoplastic effects [135,141,142,143,144,145,146,147,148]. Other examples include anti-angiogenic therapy with monoclonal antibodies to block angiogenic receptor binding and thus abrogate GBM tumor vascularization, epigenetic therapy involving histone deacetylase and methyltransferase inhibitors, oncolytic virus therapy with oncolytic viruses specific for entering followed by replicating in and then killing GBM tumor cells, and gene therapy wherein therapeutic genes or modifications of GBM tumorigenesis-related genes are introduced into tumor cells.

A long-studied but only recently resurged GBM treatment modality is boron neutron capture therapy, or BNCT [149]. First conceived in 1936, BNCT is a dual-arm therapy for several cancer types where a boron compound is infused into patients intravenously, followed by neutron irradiation of the tumor site. The “capture” or uptake of a neutron by boron-10 (a normally stable isotope) forms boron-11, which is now unstable and decays into three parts: an alpha particle (helium-4), lithium-7, and gamma rays. Theoretically, because the release radius (path length) of alpha particles and lithium-7 is about the size of a cell (5–9 µm, or microns), only tumor cells that have taken up boron-10 (and no or at least only a few normal healthy cells) will experience the major effects of BNCT (i.e., DNA damage) and ideally die because of them [135]. BNCT treatment of GBM and other gliomas have shown some promise in clinical trials, but the results are mixed, or at most comparable to or only slightly better than the standard of care [150,151]. Two key barriers to better efficacy (which result in systemic toxicity and off-target necrosis) have already been discussed earlier: the penetration of the BBB by boron-containing compounds and their tumor-specific accumulation. Interestingly, BNCT has been shown in a previous case to prolong a GBM patient’s life very significantly to approximately five years after initial diagnosis while affording excellent ability to perform activities of daily living independently [151]. Although the tumor did ultimately recur, this case study points to the high potential for BNCT to extend GBM patients’ lives without its side-effects significantly reducing quality of life.

## 8. GNPs as Glioma Therapeutics

Although none have been clinically approved due to mixed or insignificant results, several studies have investigated GNP formulations for their potential as glioma therapeutics [152,153,154,155]. For example, 11–20 nm nanospheres were injected intravenously into an orthotopic Tu-2449 glioma mouse model and evaluated for pharmacokinetics and pharmacodynamics [156]. In this study, GNPs were found to spread throughout the tumor mass and, following irradiation, prolonged long-term survival by 36% compared to control (radiation only). In another study using 3 nm and 18 nm GNPs to compare the effect of renal clearance on orthotopic murine glioma targeting and therapeutic efficacy/biosafety, the authors found that their 3 nm GNPs experienced longer retention time in circulation, although its intratumoral accumulation was significantly less than the 18 nm GNP [157]. Furthermore, the work of Allen and O’Toole loaded the cancer-specific antisense oligonucleotide AS1411 and polyethylene glycol (PEG) on 4 nm GNP nanospheres, then tested their anti-proliferative effects on the GBM cell line U87MG [153]. The authors found that the optimized GNPs synthesized at 1 μM, 2 μM, and 5 μM all caused significant decreases in proliferation, although no in vivo experiments were performed. Kumthekar and Stegh published in 2021 on a human phase 0 clinical trial of an RNAi-based GNP treatment of patients with recurrent GBM who then underwent tumor resection [154]. Importantly, no patient toxicity was observed, GNP pharmacokinetics indicated accumulation in tumor-associated tissues and cells, and a correlation with reduction in the targeted GBM oncogene expression was seen.

In 2019, Zhao and Zhao reported their GBM-specific peptide chlorotoxin-functionalized polymer-entrapped GNPs for SPECT/CT imaging and radionuclide therapy [152]. The authors found throughout in vitro and in vivo experiments that, following radioiodine labeling, their ~151 nm GNPs served as an efficient nanoprobe for targeted SPECT/CT imaging and radionuclide therapy of glioma (across cell lines, a subcutaneous tumor model, and a rat intracranial glioma model). Lastly, Yu and Xu reported in 2022 on their GNP conjugated to TMZ for mediating photothermal therapy of a drug-resistant GBM [155]. Targeted to GBM (which overexpress the ephrin type-A receptor 3 (EphA3)) by an anti-EphA3 antibody, GBM cells and a subcutaneous GBM mice model treated with these ~46 nm GNPs were irradiated with an NIR laser and studied for their apoptosis induction, resistance mechanism, and antiglioma effects. The authors found that the resulting heat produced was tumoricidal in vitro, that GNPs were synergistic in combination with TMZ on reducing cell viability, that cell cycle arrest and apoptosis were concomitant with one another following treatment, that p53 expression may be activated following treatment, that treatment had significant antiglioma effects on GBM subcutaneous tumors and caused significant elevated temperatures in the mice glioma model, and that the rat intracranial tumor model also exhibited increased brain temperature post-irradiation which subsided to baseline over time together with GNP concentration.

Taken together, these studies suggest that the advantageous properties of GNPs for treating gliomas like GBM have strong potential to come into play as clinical therapies [129]. With increasing GBM resistance to chemotherapy and development of delivery strategies, nanomedicine can be further optimized to achieve more effective GBM treatment, including potentially as a delivery vehicle for boron [135]. With their many favorable characteristics, easy fabrication, and relative biosafety, GNPs are an ideal drug delivery vehicle for a wide array of ligands and therapeutics/theranostics, including boron-containing compounds for BNCT (Figure 4). Thus, we envision that GNPs are the future of GBM therapeutics/theranostics research and development.

## 9. GNP Therapeutic Biosafety

GNP synthesis requires the use of potentially cytotoxic chemicals and compounds. Therefore, standardized removal/filtering/dialysis of these unreacted or excess chemicals and compounds is needed during drug development and especially before preclinical animal and finally clinical studies. For example, two independent studies in 2006 [35] and 2010 [158] evaluated the toxicity of the cationic surfactant cetyltrimethylammonium bromide (CTAB)-coated GNPs. Chithrani and colleagues in 2006 coated nanorods with the CTAB [35]. This positively charged coat makes the GNPs prone to “sticking” to the negatively charged cell membranes of human cells [29]. Therefore, before exposing the HeLa cell line to nanorods of varying size, the study authors importantly replaced the CTAB passivating coat with citric acid by chemical exchange as a ligand. The Chithrani paper found no indication of toxicity (measured by cell viability) in the HeLa cells exposed to the citrated nanorod formulations. Despite CTAB’s stickiness, twin studies published in 2010 by Bartneck and colleagues demonstrated that neither CTAB-passivated nor polyethylene oxide (PEO) polymer-passivated GNPs reduced cell viability when incubated with primary human blood phagocytes in both the short-term (2 h) and the long-term (7 days) [158].

As was true for GNP properties, GNP biosafety strongly hinges on their size and surface ligands/coating. For example, gold clusters/nanoclusters and nanodots, varying in size from subnanometers to 20 nm, also vary widely in their respective consensus definition across the field of synthetic chemistry [159]. However, the work of Tsoli and colleagues in 2005 on the cell culture of eleven cell lines supplemented with gold cluster GNPs resulted in varying degrees of sensitivity to the treatment [160]. The metastatic melanoma cell line BLM demonstrated a dose-dependent reduction in cell viability after 72 hr gold cluster treatment. Here, 0.4 µM gold clusters were sufficient for 100% cell death, whereas the cisplatin-treated group still exhibited about 90% viability. After analyzing the intracellular distribution of the clusters, the group hypothesized that the gold cluster binds to the major groove of DNA selectively and stably, and thus poses a potentially potent transcriptional block reflected by their data. Pernodet and colleagues showed that incubation of dermal fibroblasts with ~13 nm citrated gold nanoparticles for six days at 0.1 and 0.6 mg/mL resulted in major effects on cell morphology [161]. These effects included the narrowing of the cell footprint (and thus less cell adhesion) and reduced proliferation (related to shrinking of actin fiber diameter and formation of actin “dots” instead of normal stress fibers, which point to depolymerization and/or actin-fiber modulation). The group performed a GNP treatment time-course with TEM, finding that three hours was the minimum time for GNPs to be found in cellular images, specifically in vacuoles. In terms of additional mechanisms of cytotoxicity, after six days of incubation, these vacuoles became pervasive and extremely large to the point of enlarging the cells and disturbing normal cell structure and function (including protein production) as a result. 

In sharp contrast, Shukla and colleagues found that gold nanoparticles from 3 to 8 nm in size incubated with a macrophage cell line for 3–72 h are not cytotoxic, reduce reactive oxygen species (ROS) and nitrite species, and do not induce proinflammatory cytokine production [162]. Connor and colleagues investigated GNPs ranging in size from 4 to 18 nm and capping agent (cysteine, citrate, glucose, and CTAB), finding that endocytosis by a leukemia cell line did not result in any apparent cytotoxicity measured by MTT and proliferation assays at GNP concentrations of up to 25 µM, over 2–5 days [163].

That said, surface coatings and free unbound ligands are a potential source of cytotoxicity if GNPs are not prepared properly. In a related work, Goodman and colleagues provided evidence for the moderate toxicity of cationic GNPs and low to no toxicity of anionic GNPs (the cores of both being 2 nm in size) in fibroblast-like and red blood cells (as well as interestingly in the bacteria model organism *E. coli*) over the course of 1–24 h [164]. The work of Pan and colleagues used gold nanocluster treatments (ranging in size from 0.8 to 15 nm) to test GNP cytotoxicity in the following cell types: epithelial and endothelial cells, macrophages, and connective tissue fibroblasts [159]. When the cancer cell lines among the cell types listed above were exposed to the 1.4 nm nanoclusters for one hour, membrane blebbing and vesicle formation accompanied swelling and loss of adherence. These effects worsened at the 12 h timepoints into strong evidence of apoptosis and secondary necrosis. The 48 h and 72 h treatments demonstrated clear and significant reductions in viability, but importantly only for the nanocluster-sized (1–2 nm) GNP preparations since their effects were compared to the 15 nm GNP preparations and untreated control cultures.

Although larger GNPs possibly have improved BBB uptake because of active transport mechanisms, they also stay longer in circulation and preferentially accumulate in the liver and/or spleen, posing possible toxicity risks [165]. Interestingly, the study of Ding and colleagues importantly demonstrated no measurable negative effects of the aptamer-functionalized Apt-Au@MSL treatment in the nude mice tumor xenograft model, despite the very large (150 nm) core and hydrodynamic sizes of the final GNP [166]. Of note, both the anti-cancer effects and toxicity portions of the study evaluated these aspects 24 and 30 days post-treatment, indicating at least acute biosafety in vivo.

## 10. Conclusions

Gold nanoparticles (GNPs) are widely applied in the optical and biomedical industries. Because of their relative biosafety, well-defined synthesis methods, and favorable physicochemical properties, GNPs hold immense promise across the sciences, especially for cancer therapeutics/theranostics. Their potential applications for treatment of the malignant glioma Glioblastoma multiforme (GBM) are manifold. The extremely poor prognosis, because of the inevitable relapse of GBM despite the standard of care, requires continued intensive research into alternative therapies. Of the many therapies investigated, GNPs hold many advantages and few downsides, particularly in their potential for mediating boron neutron capture therapy (BNCT). Given the tumor-targeting issues of current boron carriers, we propose that using GNPs as delivery vehicles/platforms for GBM therapeutics/theranostics has excellent potential for the successful development of a clinical treatment. We especially point to GNPs as a delivery vehicle for boron-10-containing compounds to mediate more effective and biosafe BNCT in glioma/GBM patients.

## Figures and Tables

**Figure 1 materials-17-01153-f001:**
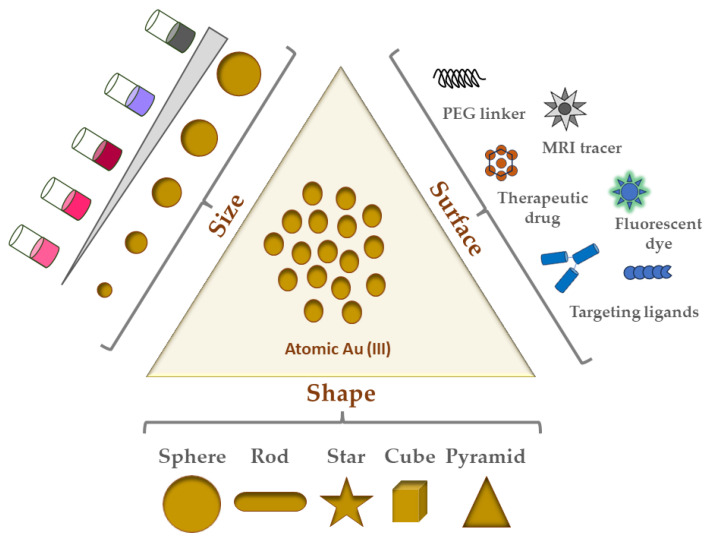
Illustration of GNPs applied in cancer therapy and their functionalization.

**Figure 2 materials-17-01153-f002:**
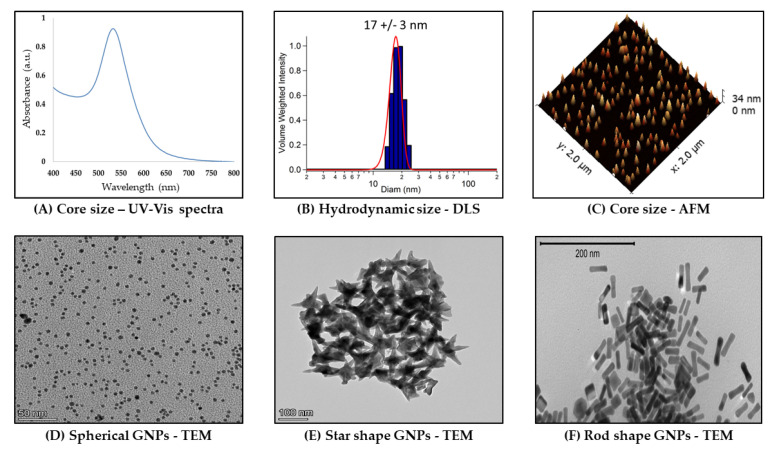
Commonly used techniques for physicochemical characterization of GNPs. The red line in panel (**B**) is a Gaussian fit to the hydrodynamic size distribution.

**Figure 3 materials-17-01153-f003:**
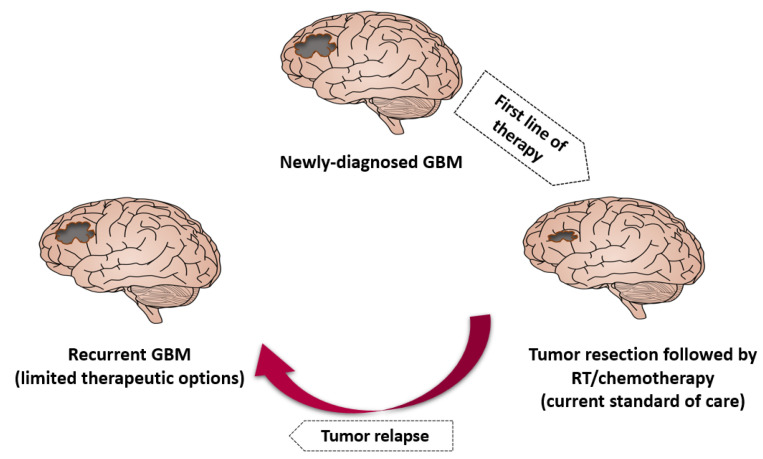
Illustration of GBM standard of care and recurrence.

**Figure 4 materials-17-01153-f004:**
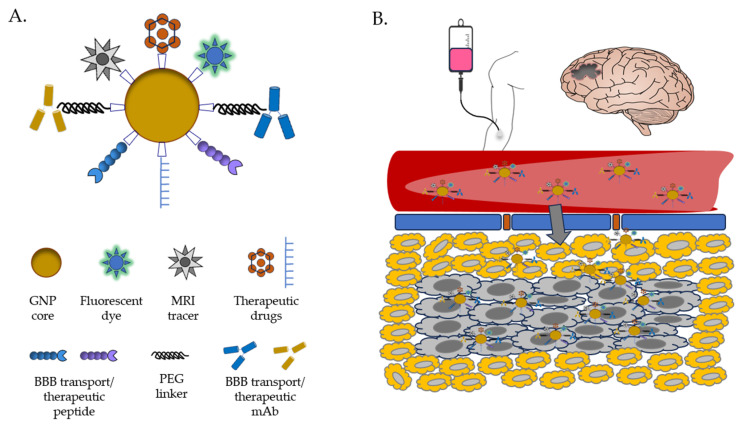
Schematic presentation of GNP for tumor targeting and BBB delivery. (**A**) Multifunctional GNPs illustrating potential surface modifications. (**B**) Illustration of drug delivery across the BBB.

**Table 1 materials-17-01153-t001:** Effect of various shapes and sizes of targeted gold nanoparticles in tumor treatment.

Shape (Size nm)	Surface Coating	Application	Reference
Nanosphere (~35)	Cyclic RGD peptides	Sarcoma treatment	[87]
Nanosphere (89.7 ± 2)	Aptamer-conjugated PTX (Paclitaxel)	Breast cancer	[88]
Nanosphere (18.5 ± 2)	Silica coating	Ductal mammary carcinoma	[89]
Nanosphere (80)	EGF peptides, Doxorubicin	Brain cancer	[90]
Nanosphere (26.5 ± 1.1)	Prostate-specific membrane antigen (PSMA-1)	Prostate cancer	[91]
Nanosphere (62.9 ± 0.7)	small-interfering RNA (siRNA) & RGD peptide	Lung cancer	[92]
Nanosphere (56.37 ± 3.04) Rods (22.41 ± 1.01)	Cyclic RGD peptides	Non-small cell lung cancer (NSCLC)	[93]
Nanostar (78.0 ±2.9)	Chlorin e6-func-tionalized	Breast cancer	[94]
Nanosphere (29 ± 2)	Doxorubicin or Varlitinib and PEG	Pancreatic Adenocarcinoma	[95]

**Table 2 materials-17-01153-t002:** Selected publications on targeted gold nanoparticle drug delivery and their biosafety. ND = not determined.

Shape/(Size nm)	Ligand(s) and Application	Biosafety	Reference
Nanosphere (10–12)	Gentamicin as an antibacterial for *S. aureus* infection	ND	[106]
Nanosphere (~13.2)	Bromelain (BRN) and levofloxacin as antibiotics for treating *S. aureus* and *E. coli*	ND	[107]
Nanosphere (20–40)	Thiolated antisense oligos for restoring cardiac function in postmenopausal diabetic mice	No adverse effects seen after 1 h up to 4 weeks	[108]
Nanosphere (20–30)	Resveratrol and Tc-99m radiolabel for treating colon tumor rat model	Low red blood cell hemolysis. High tumor: normal ratio.	[101]
Nanosphere (15)	Cyclic PEG for enhancing colloidal stability as well as cellular uptake/tumor accumulation	Slow blood clearance, low accumulation in colon tumor	[109]
Nanosphere (~21 ± 10)	Dextran as a capping agent for enhanced colloidal stability	No adverse effects seen after 24 h, 7 days, or 14 days	[111]
Nanorods (20–100)	CTAB surfactant, PAH/PAA polymer for accumulating in and inhibiting colon carcinoma	Polymer coating non-inhibitory, CTAB growth-inhibitory	[116]

## Data Availability

Not applicable.

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
