# Peer review of "Designing Gold Nanoparticles for Precise Glioma Treatment: Challenges and Alternatives"

_materials, 2024, doi:10.3390/ma17051153_

Round 1

Reviewer 1 Report

Comments and Suggestions for Authors

This review article examines the use of Gold Nanoparticles for Glioma Treatment. Despite the fact that in many places in the article (in the title, in the abstract, in the introduction and conclusion) the authors indicate that 'This review discusses historical and current GNP therapeutic applications as a drug delivery vehicle for treating glioma', data on the use of Gold Nanoparticles for Glioma Treatment are practically absent. The article consists of 8 sections: the first 7 sections describe Gold Nanoparticles, their physicochemical properties and use as drug delivery systems. Section 8 is dedicated to Gliomas and touches a little on the use of nanoparticles for treating glioma. However, there is not a single link dedicated to gold nanoparticles for treating glioma. There is only 1 figure (#4) and a brief description that 'gold nanoparticles will be at the forefront of future GBM therapeutics/theranostics research and development'. Due to the lack of any information about what was supposed to be the purpose of the article, the article cannot be accepted for consideration. This article can be accepted for consideration only after thorough revision and only after that comments can be made.

In addition, it should be noted that there are a large number of articles on the use of Gold Nanoparticles in Glioma Theranostics, including a number of review articles (for example, Norouzi M. Gold Nanoparticles in Glioma Theranostics. Pharmacol Res. 2020 Jun;156:104753. doi: 10.1016/j.phrs .2020.104753. Epub 2020 Mar 21. PMID: 32209363). Authors must justify how their article differs from existing ones.

Author Response

1. Our response: Thank you for providing your feedback on our manuscript. We greatly value your suggestions and have taken them into account for the improvement of our work. Following your recommendation, we have revised Section 8.3.1 to include additional references and discussions focusing on primary literature exploring the utilization of gold nanoparticles in glioma therapeutics.

Specifically, we have integrated four new references into this section of the manuscript, which are succinctly summarized as follows:

  1. Zhao and Zhao (2019) conducted a study utilizing chlorotoxin-functionalized polymer-entrapped gold nanoparticles for SPECT/CT imaging and radionuclide therapy.
  2. Kumthekar and Stegh (2021) published a paper detailing a Phase 0 clinical trial involving gold-nanoparticle-based RNAi therapy in recurrent GBM patients.
  3. Allen and O’Toole (2022) synthesized gold nanoparticles conjugated with cancer-specific antisense oligonucleotide and polyethylene glycol (PEG) to investigate their anti-GBM effects.
  4. Yu and Xu (2022) explored the efficacy of gold nanoparticles conjugated with anti-EphA3 antibody and temozolomide in photothermal therapy for GBM/gliomas.

The inclusion of these additional references and discussions enhances significantly the comprehensiveness and relevance of our manuscript to the topic of gold nanoparticles for glioma treatment. These revisions address the recommendations made by the reviewer and greatly strengthen our manuscript.

2. 

Our response: Thank you for your comment. Gold nanoparticles have gained broad popularity in many research fields and thus are commonly studied and reviewed in the literature. While many other reviews, including the one pointed out by the reviewer above, focus on gold nanoparticles in theranostics. The theranostic approach generally focuses on imaging and treatment using a single platform. The scope of our article focuses on multiple aspects of gold nanoparticles including but not limited to GNP types, shapes, size, physicochemical properties, and therapeutic applications for the central nervous system, and a brief for a few other diseases.

Reviewer 2 Report

Comments and Suggestions for Authors

The current manuscript 'Designing Gold Nanoparticles for Precise Glioma Treatment: Challenges and Alternatives' discusses the application of GNPs for Glioma treatment. However, the manuscript is not presented in an organized way and can be improved as below:

1) The alternatives treatments were not discussed besides the BNCT. How about other nanoparticles or stimulus receptive materials? A thorough discussion is needed to review the alternatives.

2)The section, 'Physicochemical characterization techniques for GNPs' is irrelevant to the main review focus.

3) More discussion in table form is needed for the section 6.2 GNP applications in cancer therapeutics; 7. GNPs as a drug delivery vehicle; 8. Gliomas, glioblastoma multiforme (GBM), and the promise of nanoparticles

4) The statement "Thus, we propose that GNPs be at the forefront of 669 future GBM therapeutics/theranostics research and development" on page 15 looks like a proposal or thesis statement, not review article.

5) Section 8. Gliomas, glioblastoma multiforme (GBM), and the promise of nanoparticles can be discussed after the introduction.

Author Response

Here we have given one-by-one answers to reviewer questions and have added their question too for better understanding. Thank you!

Our response: We thank the reviewer for insightful suggestions regarding our manuscript. Detailed response is given below:

The alternatives treatments were not discussed besides the BNCT: In the revised manuscript, we have added a broad discussion focusing on alternative treatments such as immunotherapy, immune checkpoint blockade, co-administration of convection-enhanced delivery of temozolomide and whole-cell immunization with irradiation-inactivated tumor cells. We also included sex-specific differences in immunosuppressive GBM tumor microenvironments, and viral immunization to instigate antineoplastic immune response have also all been investigated with promising preclinical results. Other therapies discussed include antibody-based anti-angiogenic therapies, epigenetic therapy, oncolytic viral therapy, and gene therapy for introducing or modifying tumorigenesis-related genes.

How about other nanoparticles or stimulus receptive materials? To keep the discussion within the scope of this article we focused only on GNPs.

A thorough discussion is needed to review the alternatives. We have added and thoroughly discussed the alternatives and the new text is highlighted in the revised manuscript.

  1. The section, 'Physicochemical characterization techniques for GNPs' is irrelevant to the main review focus.

Our response: Thank you for the comment. The physicochemical properties of gold nanoparticles determine their suitability as drug-delivery vehicles. Keeping with the scope of the article, understanding properties (size, shape, surface charge, surface functionalization, etc.) is critical for designing optimum-functioning nanoparticles for specific therapeutic applications. Moreover, this would facilitate interdisciplinary readership.

  1. More discussion in table form is needed for the section 6.2 GNP applications in cancer therapeutics; 7. GNPs as a drug delivery vehicle; 8. Gliomas, glioblastoma multiforme (GBM), and the promise of nanoparticles.

Our response: Thank you for this valuable suggestion. We have added a new Table 2 in Section 7.1 on targeted gold nanoparticle drug delivery and biosafety as recommended. Examples include gold nanospheres conjugated to antisense oligonucleotides used for restoring cardiac function in a postmenopausal diabetic mouse model, gold nanospheres adsorbed with cyclic polyethylene glycol polymer for prolonged retention in a mouse colon tumor model, and CTAB and polymer-coated nanorods for accumulating in and inhibiting the growth of human colon carcinoma cells.

  1. The statement "Thus, we propose that GNPs be at the forefront of 669 future GBM therapeutics/theranostics research and development" on page 15 looks like a proposal or thesis statement, not review article.

Our response: We thank you for the comment. As recommended, we have revised the language of the statement (highlighted in yellow color, Section 8.3.1 of the manuscript).

  1. Section 8. Gliomas, glioblastoma multiforme (GBM), and the promise of nanoparticles can be discussed after the introduction.

Our response: Thank you for the comment. We structured the manuscript by providing an overview and detailed information on gold nanoparticles and their applications, followed by connecting their potential utility to GBM therapeutics.

Reviewer 3 Report

Comments and Suggestions for Authors

This review paper discusses the use of gold nanoparticles (GNPs) as a drug delivery vehicle for treating glioblastoma multiforme (GBM). It clearly summarizes the physicochemical properties and characterization techniques of GNPs. The paper was well written and will be beneficial to those who are interested in this field.

-Could you please elaborate more on the limitations of GNPs drug delivery system and what are the challenges.

-Section 7.1 “Strategies of GNP drug delivery in the brain” and 7.2 “Examples of GNP drug delivery in the brain” focuses on the same topic. It might be better to combine them into one section.

-Is there any other GNP drug delivery application other than to the brain?

-The first paragraph of section 8.3 “Treatment modalities for glioma” seems to be overlapping with section 8.2. It might be better to rearrange them and have section 8.3 focusing on the GNPs for GBM treatment. This will lead to a better flow and make it easier to follow.

-Figure 4 summarizes the content and might be used as a graphical abstract.

Author Response

Q1. This review paper discusses the use of gold nanoparticles (GNPs) as a drug delivery vehicle for treating glioblastoma multiforme (GBM). It clearly summarizes the physicochemical properties and characterization techniques of GNPs. The paper was well-written and will be beneficial to those who are interested in this field.

Could you please elaborate more on the limitations of GNPs drug delivery system and what are the challenges.

Our response: We appreciate your highly supportive comments. We have followed the reviewer’s suggestions regarding our manuscript and have revised Section 7 as recommended.

Q2. Section 7.1 “Strategies of GNP drug delivery in the brain” and 7.2 “Examples of GNP drug delivery in the brain” focuses on the same topic. It might be better to combine them into one section.

Our response: Thank you for the recommendations. We have revised Section 7 to merge Sections 7.1 and 7.2 as recommended. It is highlighted in yellow color for your kind attention.

Q3.Is there any other GNP drug delivery application other than to the brain?

Our response: GNP drug delivery is widely-used in many other areas of research with robust applications. Considering the scope of this article, we focused on and discussed GNP applications to the brain. Nonetheless, we have also discussed applications of GNP for other cancers. Please refer to Section 6.2, and Table-2.

Q4. The first paragraph of section 8.3 “Treatment modalities for glioma” seems to be overlapping with section 8.2. It might be better to rearrange them and have section 8.3 focusing on the GNPs for GBM treatment. This will lead to a better flow and make it easier to follow.

Our response: Section 8.2 reviews the current standard of care and its inadequacies, while Section 8.3 focuses on current/recent investigations into GNPs and other alternatives for GBM treatment beyond the standard of care. We think this order could present a good clarity. Although we appreciate your suggestion.

  1. Figure 4 summarizes the content and might be used as a graphical abstract.

Our response: This is an excellent suggestion. We retained Figure 4 and created a new graphical abstract. The new graphical abstract is incorporated into the abstract.

Reviewer 4 Report

Comments and Suggestions for Authors

The manuscript well introduces the gold nanoparticles, including their properties, characterization methods, and their applications in the biomedical fields, which is important to the field. Therefore I would recommend that this manuscript be publishable if the author could elaborate on my concern about the gold nanoparticle stability.

In section 3.2, the author claimed that the gold nanoparticle colloid will be stable if the zeta potential is greater than +30 mV or less than -30 mV, which is incorrect. The gold nanoparticle could be extremely stable when the surface is coated with surfactant. Please cite and refer to this paper "Liu, Z.; Lanier, O.L.; Chauhan, A. Poly (Vinyl Alcohol) Assisted Synthesis and Anti-Solvent Precipitation of Gold Nanoparticles. Nanomaterials 202010, 2359. https://doi.org/10.3390/nano10122359". In this paper, researchers introduced PVA as a stabilizer for gold nanoparticles, which have a less zeta potential, but better stability in the aqueous solution.

Author Response

Our response: We highly appreciate your positive comments and suggestions regarding our manuscript. We have revised the end of section 3.2 to better discuss zeta potential and colloidal stability as recommended and have now cited the paper you mentioned. The revised section is highlighted in yellow.

Round 2

Reviewer 2 Report

Comments and Suggestions for Authors

Authors have addressed the review reports and improved the manuscript. I recommend accepting the manuscript. 

Comments on the Quality of English Language

Minor changes are needed. 

Author Response

Reviewer 2 (date on site: 2/12/24):

  1. Authors have addressed the review reports and improved the manuscript. I recommend accepting the manuscript.  Minor changes in the quality of the English are needed.

Our response: Thank you for providing your kind feedback on our manuscript. Following your recommendation, we have extensively improved the English throughout the manuscript (highlighted yellow in the text).  One of the corresponding authors Dr. Perry, is a native English speaker and has helped us to improve the language for this manuscript.
